# Air Pollution and Mortality Impacts

Zhe Michelle Dong [1], Han Lin Shang [2,*] and Aaron Bruhn [1]

1.  Research School of Finance, Actuarial Studies and Statistics, Australian National University, Canberra, ACT 2601, Australia; zhe.michelle.dong@gmail.com (M.D.); aaron.bruhn@anu.edu.au (A.B.)
2.  Department of Actuarial Studies and Business Analytics, Macquarie University, Sydney, NSW 2109, Australia
*   Correspondence: hanlin.shang@mq.edu.au; Tel.: +61-(2)-9850-4689

**Abstract:** This study quantifies the air quality impact on population mortality from an actuarial perspective, considering implications to the industry through the application of findings. The study focuses on the increase in mortality from air quality changes due to extreme weather impacts. We conduct an empirical study using monthly Californian climate and mortality data from 1999 to 2019 to determine whether adding PM2.5 as a factor improves forecast excess mortality. Expected mortality is defined using the rolling five-year average of observed mortality for each county. We compared three statistical models, namely a Generalised Linear Model (GLM), a Generalised Additive Model (GAM), and an Extreme Gradient Boosting (XGB) regression model. We find including PM2.5 improves the performance of all three models and that the GAM performs the best in terms of predictive accuracy. Change points are also considered to determine whether significant events trigger changes in mortality over extended periods. Based on several identified change points, some wildfires trigger heightened excess mortality.

**Keywords:** climate risk; wildfires; change-point detection; Multivariate Time Series; machine learning





## 1. Introduction

The impact of climate change and extreme weather events is becoming increasingly pervasive. Extreme weather events such as bushfires and wildfires significantly impact air quality. For example, the Australian bushfires of 2019–2020 resulted in over 18 million hectares of land burned, and 57% of the total population in Australia being exposed to bushfire smoke (Borchers Arriagada et al. 2020). In particular, the UN Environmental Programme reports that Canberra specifically measured the worst air quality index of any major city in the world, redefining the concept of extremes (UNEP 2020). Nationwide, this led to an estimated over 3000 additional hospital admissions, over 1000 additional presentations of asthma, and more than 400 excess deaths over the three months (Borchers Arriagada et al. 2020).

Similarly, in the United States (US), Californian wildfires in 2018 have destroyed over 800,000 hectares of land (Bladen 2020). Wildfires have hit California, Oregon, and Washington hardest. While as a proportion of total landmass, the US impacts are relatively less than the 2019–2020 Australian bushfires, the direct impacts have been increasing. Between 1972 and 2018, there was a fivefold increase in the annual burned area, increased fatalities from wildfires, and total state spending on fire suppression of over USD $1.5 billion (Williams et al. 2019). Recent wildfires in the US have been more severe and frequent than historical fires, with research suggesting this is also an impact of climate change (Jolly et al. 2015). Under climate change scenarios (e.g., 2–3° Celsius warming scenarios), extreme weather events such as wildfires are expected to occur with greater frequency and severity (Abatzoglou et al. 2019). Thus, there is a compelling need to understand better the impacts on air pollution mortality triggered by extreme events.

Air pollution is a broad term that can refer to a mixture of gases and particles. Typically, PM2.5 (particle mass with an aerodynamic diameter less than 2.5 μm) and tropospheric

ozone are the two indicators used to quantify exposure to air pollution. PM2.5 particulate matter is small enough to enter the bloodstream via the lungs, and is produced through human (e.g., traffic, manufacturing) and natural causes (e.g., fires). PM2.5 is the most consistent and robust predictor of mortality in studies of long-term exposure (Cohen et al. 2017). Exposure to PM2.5 has been shown to trigger an increase in mortality by 6% per 10 µg/m$^3$ increase independent of age, gender, and region (Chen et al. 2008). Prolonged exposure to particulate matter PM2.5 results in elevated mortality responses. Studies in the US have revealed that 25% of all PM2.5 in the US can be attributable to wildfires (Burke et al. 2021). The Global Burden of Diseases, Injuries, and Risk Factors Study in 2015 (Global Burden of Disease 2016) estimated the burden of disease attributable to 79 risk factors in 195 countries from 1990 to 2015. It identified ambient air pollution as the fifth-ranking mortality risk factor, contributing to 7.9% of total global population mortality, and 4.2% in terms of disability-adjusted life-years (Cohen et al. 2017). In that study, ambient air pollution accounted for 7.6% of total population deaths globally (Cohen et al. 2017). The five causes of death that the GBD study identifies as being impacted by ambient air pollution are (WHO 2018):

- Acute lower respiratory infections
- Chronic Obstructive Pulmonary Disease (COPD)
- Lung Cancer
- Ischaemic Heart Disease (IHD)
- Stroke

Location-based factors can impact the quality of life through economic wellbeing and public spending, demographics, access, and quality of local resources. Cupido et al. (2020) explore the spatial distributions of mortality rates for ages 65+ by US county, and their analysis shows including spatial filters (which factors in geographic factors) on mortality improves the analysis. Another factor linked with location is air quality, with studies investigating the impact of locality on infant mortality by studying exposures over time and the resulting deaths. The most common findings for infant mortality are that CO and PM2.5 have an impact on preterm births and birthweights but with CO having a more significant impact on mortality compared to PM2.5 (Currie et al. 2009). There remains little empirical research to link extreme events impacting air quality to applied settings, such as excess mortality, from a life insurance perspective.

Yu et al. (2020) investigate the association between long-term exposure to low-level PM2.5 and mortality in Queensland, Australia, between 1998 and 2013. A difference-in-difference approach was adopted, and analysis was performed for aggregated and cause-specific deaths. This research concludes that even for PM2.5 concentrations below the current World Health Organisation annual standard (10 µg/m$^3$), there is a 2.02% increase in mortality for a 1 µg/m$^3$ increase in PM2.5. This corroborates the findings of the GBD studies and further suggests that PM2.5 exposure, even at low levels, presents additional mortality risk (Yu et al. 2020).

Further studies have been conducted in China and Germany to understand the impact of PM2.5 on mortality. In China, Cao et al. (2018) constructed a geographically weighted regression model, and findings showed a significant positive correlation between PM2.5 concentration and lung cancer mortality and that the impacts increase over time as PM2.5 concentration increases. Similarly, Breitner et al. (2009) find higher PM10 (particulate matter with a diameter less than 10 µm) concentrations result in higher mortality. As PM10 concentrations decrease, mortality is also shown to decrease, thus suggesting improvements in air quality can indeed result in improvements in mortality (Breitner et al. 2009). Furthermore, low emission zones in Germany have been shown to improve cardiovascular health by 2–3%, particularly for people aged over 65 (Margaryan 2021).

Other research has focused more on longitudinal trends; for example, Huang et al. (2019) performed a cohort study based in China to understand the impact of PM2.5 on stroke over 15 a period of years from 2000 to 2015. Participants were tracked on a number of variables: smoking, blood pressure, medical history, and lifestyle risks. Long-term exposure to ambient PM2.5 at relatively high concentrations is positively associated with

incident stroke and its major subtypes. A more recent study on stroke mortality in China corroborates these findings, with data suggesting that temperature is also a critical factor in determining mortality (Yang et al. 2021).

Behavioral responses, government regulations, socioeconomic status, and household heating are other factors that impact the mortality response to air quality. A factor that has also been considered in the literature is the impact of avoidance behavior or behavioral responses to government warnings to vulnerable groups to avoid exposure. Janke (2014) found that there is a relationship between hospital admissions in children and air pollution as measured by noxious gases in England. However, there is no statistically significant evidence to suggest that there is a health impact from avoidance behavior (Janke 2014). State policy also plays a vital role in the overall level of pollutants in the air by regulating industries and taxing carbon. Studies on government intervention in China to reduce air pollution show that there are statistically significant improvements in infant mortality from government measures that limit emissions (Tanaka 2015). This suggests that the degree of avoidance or regulation is important in understanding the overall impact of air pollution on mortality. Therefore, factors other than the ambient exposure to PM2.5 can impact mortality.

Studies on economic status and the resulting impact on children's health demonstrate that lower socioeconomic households are more sensitive to changes in PM10, as demonstrated by Jans et al. (2018). Their study uses inversion episodes to measure the impact of poor air quality on children's health through health care visits in Sweden. The finding suggests that the impact is greater on children in lower socioeconomic households due to a larger proportion of children with poorer baseline health (Jans et al. 2018). Another factor related to socioeconomic status is likely the type of heating used in households. Studies show that coal-fired heating systems indeed lead to deterioration in air quality and mortality (Cesur et al. 2018; Fan et al. 2020).

Actuarial studies have focused on the mortality risks due to property damage arising from weather events driven by climate change (Miljkovic et al. 2018), and climate-related risks on index-based insurance (McCarthy and Wang 2021). Separately, there have been studies delving into the relationship between extreme mortality and extreme temperature: Li and Tang (2022) show joint extremes in cold weather, and old-age death counts exhibit the strongest level of dependence. Thus, evidence points to weather extremes impacting mortality and life insurers' risk exposures.

In this study, we tested the impact of PM2.5 concentration on mortality and used the residual explanatory power of location and time as a proxy for other factors, including seasonality. Excess monthly mortality is defined as the increase in mortality for a particular month from the average mortality observed over the past five-year period. The UK Office of National Statistics has adopted similar approaches to analyzing mortality experience (Office for National Statistics 2021) and other research to understand the excess mortality impact of COVID-19 (Shang and Xu 2022), where observed historical mortality was used to determine the expected mortality. Knowing that there is an impact on mortality from air pollution, we address whether changes in PM2.5 concentration affect excess mortality.

To better understand the above, we analyze data from 1999 to 2019 on mortality and PM2.5 concentration readings for counties in California. This paper contributes to the existing literature in understanding the climate impact on population mortality by:

- Developing a gradient boosting regression model to understand and predict the significance of the relationship between air quality and excess mortality over time. Comparisons are performed between the proposed machine learning approach and established regression methods. We find that PM2.5 concentration is a significant factor in determining excess mortality and that a machine learning approach does not always perform better than established methods.
- Understand the impact of extreme weather events by applying change point analysis to understand the regression models better and improve predictive power. We assess

the identified change points against known events and find that certain wildfires trigger structural changes in excess mortality.

Based on existing literature, there is a link between mortality risk, climate risk, and air quality, as measured by PM2.5. Thus, this research links the medical and longitudinal observations around air quality impacts on health with mortality risk. Section 2 details the data sources and Section 3 expands on the methodology and results. Section 4 considers change point analysis in the empirical data, and Section 5 discusses the results of the analysis, the implications of these findings, the limitations of this study, and the applications.

## 2. Data

Historical data are sourced from the Environmental Protection Agency in the US ("EPA") on PM2.5 and Wonder CDC (Centers for Disease Control and Prevention, National Center for Health Statistics 2020) on mortality to assess the impact of air pollution on mortality in California from 1999 to 2019. We then overlay the results with historical temperature data and indicators of wildfire events from the California Department of Forestry and Fire Protection to investigate the impact of wildfire events on population mortality.

Monthly aggregated death statistics by age, gender, and county is used to calculate the expected mortality, computed as a rolling 5-year average death rate per mille (i.e., per thousand lives). The rolling 5-year average is the average of the previous five years of data. The variable of interest is excess mortality or the difference in actual observed death rates from the expected mortality. For years 1999–2004, the expected mortality is calculated by taking the rolling average of all months of data available, up to 5 years.

Our empirical study uses four types of data sources: PM2.5 concentrations, temperature indices, historical wildfire indicators for California, and actual monthly mortality data by age, gender, and county.

### 2.1. PM2.5 Concentrations

Daily PM2.5 concentration by US county over 1999 to 2019 was sourced from the EPA (URL: https://www.epa.gov/outdoor-air-quality-data/download-daily-data, accessed on 30 September 2021) and Wonder CDC (URL: https://wonder.cdc.gov/controller/datarequest/D76, accessed on 30 September 2021).

Readings are available on daily and monthly bases. In our analysis, we use exposure weighted the concentration of PM2.5 over 24 h to determine an average, maximum, and minimum PM2.5.

Figure 1 shows average annual PM2.5 concentrations for monthly maximum, average, and minimum daily readings across counties in California. The solid graph represents the summarised PM2.5 concentrations (annual mean, maximum, minimum). The vertical bars show the variability defined by the standard deviation of monthly PM2.5 concentrations each year. The standard deviation is the biggest for maximum PM2.5 concentration, compared to average or minimum. In more recent years, the variability of PM2.5 concentrations has been increasing, which is explored further in the change point analysis in Section 4.

### 2.2. Historical Wildfire Indicators

Historical wildfire records are sourced from the California Department of Forestry and Fire Protection. Information on historical wildfires and temperatures in the same region is also included in the analysis. We show the average mean, minimum, and maximum PM2.5 concentrations across all Californian counties and the total wildfires indicators (by deadliest, destructive, and size indicators) over time. This is used to indicate whether there is a causal relationship between wildfire events (as identified by the California Department of Forestry and Fire Protection) and increases in PM2.5 concentrations.

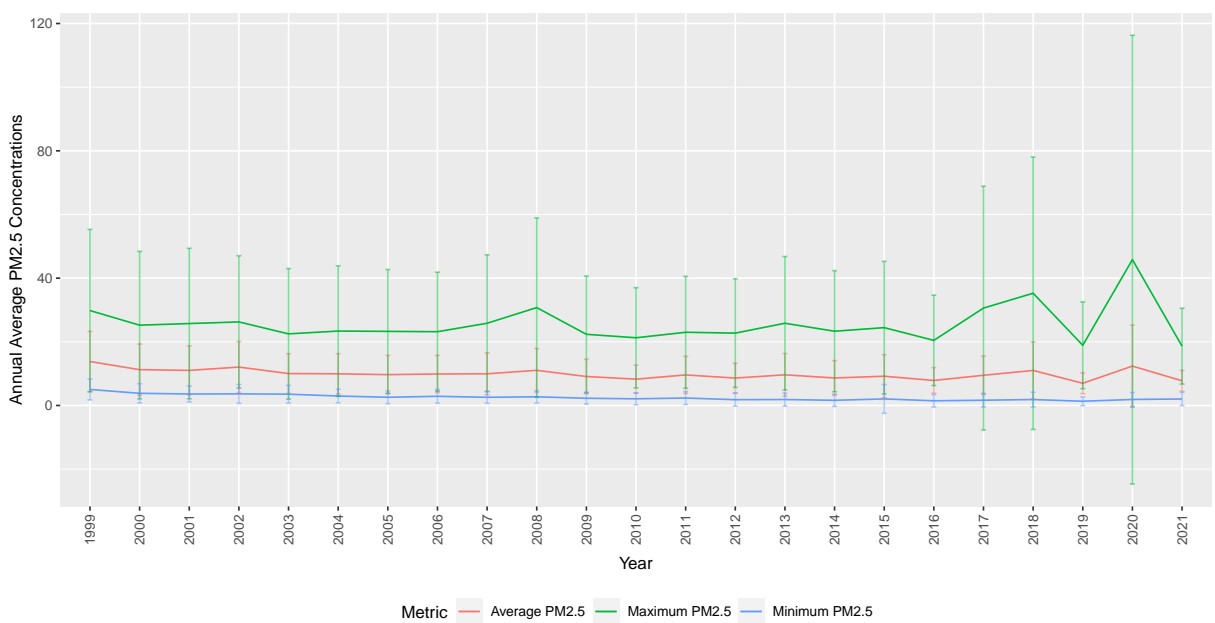

**Figure 1.** Average PM2.5 in California.

Our investigations find spikes due to large wildfires at a county-specific level. PM2.5 concentrations are not perfectly aligned with wildfire events when averaged across all counties due to the non-uniform distribution of monitoring stations and meteorological factors, including wind. To show spikes in PM2.5 concentration aligned with large wildfire events, Figure 2 focuses on one county (San Diego). Increases in PM2.5 concentrations due to wildfires in a particular county would be less obvious. It would be offset by more typical levels of PM2.5 readings in other counties further from the fire. The model compares the effects, including county-level distinctions, thus accurately depicting the relationship between PM2.5 concentrations and wildfires.

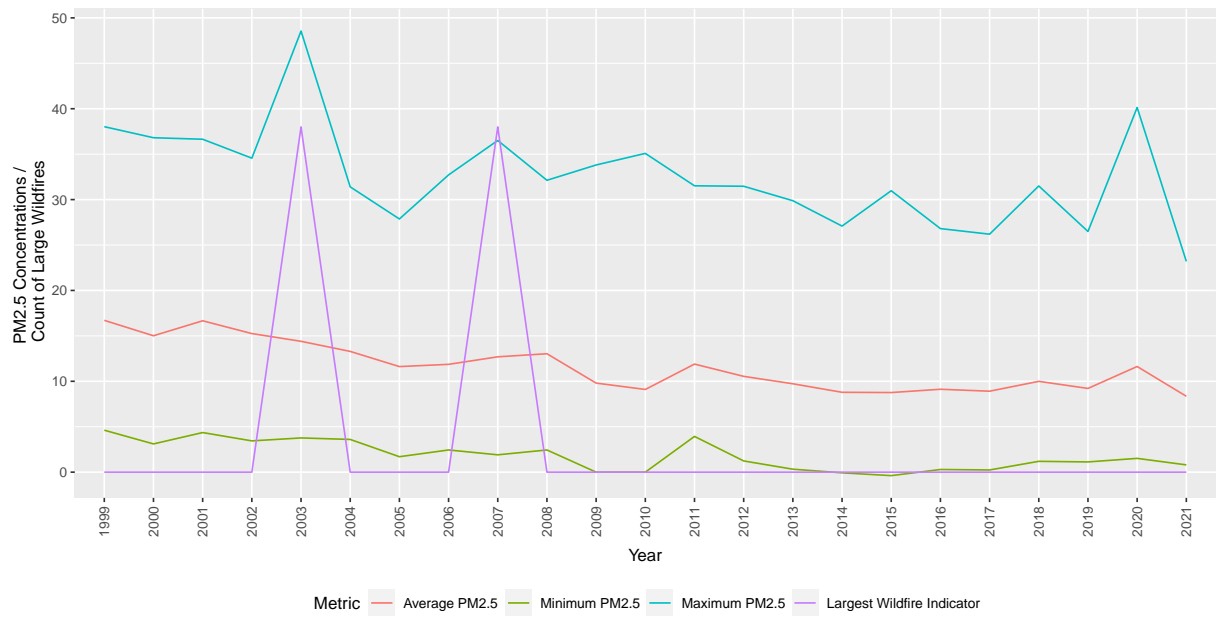

**Figure 2.** Relationship between PM2.5 concentration readings and wildfire events: San Diego County Only.

*2.3. Temperature Indices*

We include a metric on the temperature in our analysis given the known link between extremes in temperature and mortality (Li and Tang 2022).

Data on temperature indices were sourced from the Actuaries Climate Index, specifically the extreme temperature measures reported through this index. The Actuaries Climate Index was developed as a monitoring tool in collaboration between the North American actuarial organisations: the Canadian Institute of Actuaries, the Society of Actuaries, the Casualty Actuarial Society, and the American Academy of Actuaries (Actuaries Climate Index 2019). The index was developed using 1960–1990 as reference years, and subsequent reporting is based on this standardised timeframe. For analysis, the Actuaries Climate Index output on temperature indices was used from 1999 to 2021.

The index has six components, including:

(1)     Frequency of temperatures above the 90th percentile (T90)
(2)     Frequency of temperatures below the 10th percentile (T10)
(3)     Maximum rainfall per month in five consecutive days (P)
(4)     Annual maximum consecutive dry days (D)
(5)     Frequency of wind speed above the 90th percentile (W)
(6)     Sea level changes

The Actuaries Climate Index (ACI) (Actuaries Climate Index 2019) components are constructed in a uniform grid across the USA and Canada. There are 12 regions defined in the ACI. The ACI region of interest is Southwest Pacific ("SWP"), as it captures the state of California as well as other states (Arizona, Colorado, New Mexico, Nevada, and Utah).

For this analysis, in Figure 3, the focus is on T90 and T10 metrics for the Southwest Pacific (SWP) region where California is located. Data from the ACI are provided monthly, seasonally, standardised, and smoothed. In our preliminary analysis, the unsmoothed and unstandardised measures of T90 and T10 appear the most suitable to understand the relationship between PM2.5 and mortality.

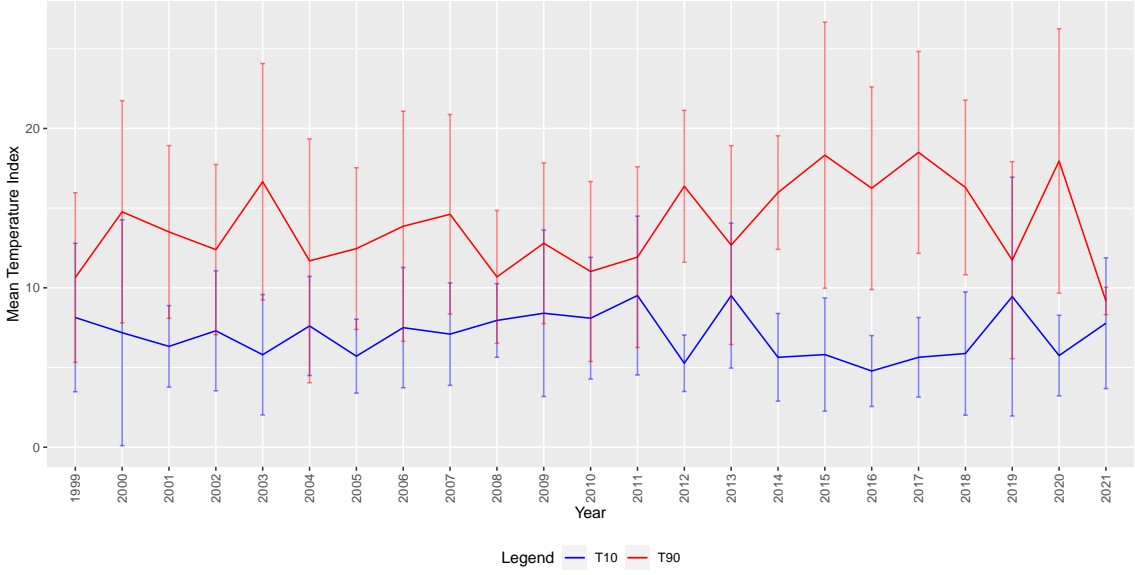

**Figure 3.** A plot of T90 and T10 for the Southwest Pacific Region over the period 1999 to 2021.

*2.4. Mortality Data*

Actual mortality data over the investigation period (1999–2019) are from the Wonder CDC. For analysis, aggregated deaths in California by county, gender, and five-year age bands were used.

Expected deaths for each month are defined as the rolling average of actual deaths (by age, gender, and county) over the previous five years. Since the data are in monthly timesteps, this is expressed as:

$$Expected Mortality_{i,t} = \frac{\sum_{n=t-60}^{t-1} Actual Mortality_{i,n}}{60}$$

where *i* represents the mortality rate by age, gender, county cohorts.

Similar methods have been adopted by recent analysis to understand the excess mortality impact of COVID-19 Shang and Xu (2022), and through regular mortality statistics reporting by the UK Office of National Statistics (Office for National Statistics 2021). In COVID-19 analysis by Shang and Xu (2022), excess deaths were analyzed by comparing observed weekly deaths by age group and gender throughout 2020 to values from the prepandemic period to perform change point analysis further and understand structural breaks in the data. By adopting this approach to define expected deaths and excess deaths to understand climate risk and air pollution, the expected mortality will capture historical impacts of PM2.5 and other changes in mortality over time (e.g., economic trends and trends in smoker's habits). This definition also more closely reflects the industry practice of setting expected mortality assumptions based on historical experience. It is sufficiently credible and better understands excess deaths and structural breaks.

Population estimates are sourced from US Census estimates by age, gender, county, and year to calculate the death rate. The response variable of interest is the excess deaths per mille (per thousand lives). In Figure 4, the excess deaths are defined as the difference between the observed deaths per month.

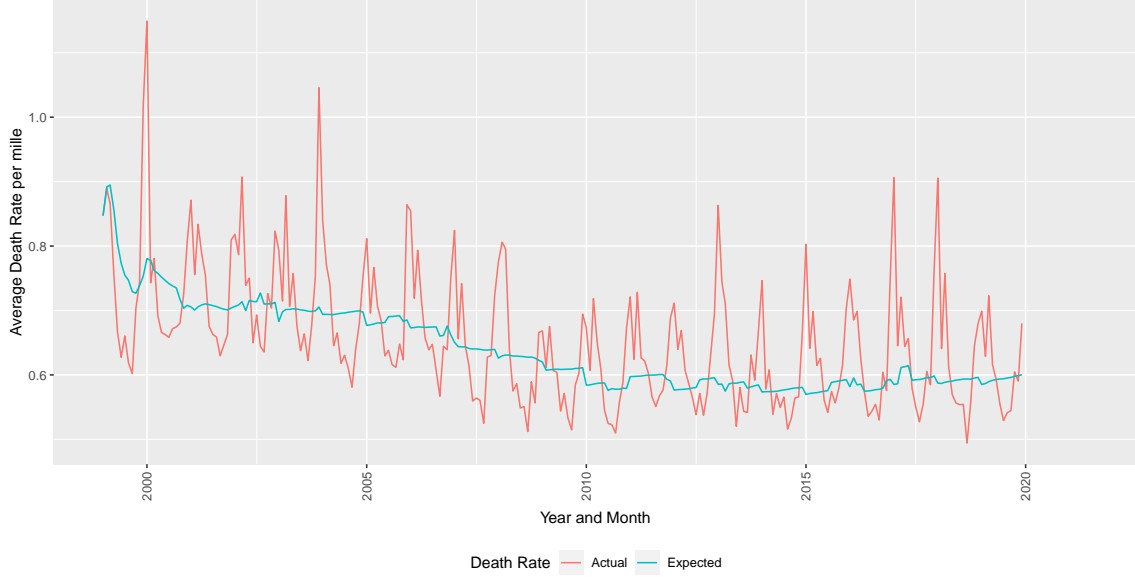

**Figure 4.** Death rate by year and month: Actual and expected.

## 3. Methodology and Results

Three approaches are compared: we begin by considering a linear regression and then compare the fit with GAM and XGB regression. A generalised linear model is a flexible generalisation of linear regression, and it allows the response variable to be related to explanatory variables via a link function. The response variable in this case is the excess mortality rate. The explanatory variables are:

- maximum PM2.5 concentration (with and without lags)
- T90 and T10 indices represent the frequency of temperatures above the 90th percentile and below the 10th percentile for the southwest Pacific region
- five-year age band, and gender
- indicator of whether that particular month had deadly, destructive, or largest fires as classified by the California Department of Forestry and Fire Protection
- month and year
- county

Preliminary analysis shows that average and minimum PM2.5 concentrations similar to but less significant than maximum PM2.5 concentration readings. Therefore, we show

results using monthly maximum PM2.5 readings. To assess finite-sample performance, we compare each model's fit and factor significance. Akaike Information Criterion (AIC) is used as a selection criterion for the LM and GAM models. For comparing accuracy in predictions, Root Mean Squared Error (RMSE) is used. Analysis of Variance was run in R to compare the two models and concluded the GAM performed better (R Core Team 2022).

The RMSE is the mean error between the predicted and measured values. It is one of the most frequently used metrics to assess the general performance of the prediction models. Therefore the closer RMSE is to 0, the better the predictive model.

The linear regression result indicates that maximum PM2.5 concentrations are a significant factor in explaining excess deaths, in addition to age, gender, county, and high temperatures. While the model's overall fit is poor, the RMSE relative to a random walk model is marginally better. Including the autoregressive factor improves the overall performance. The PM2.5 factor is significant, and results suggest that the PM2.5 variable is as important as age band and gender in explaining excess mortality.

The last column of Table 1 incorporates a lag of 1 month to the response variable. This is to capture any lagged effects of air pollution and extreme events on deaths based on the literature. There is evidence of a lag effect from increases in PM2.5 to morbidity responses. Results from Huang et al. (2019) and Yang et al. (2021) suggest there are long-term effects from prolonged exposure to PM2.5. The literature suggests that there are short-term lag effects for spikes in PM2.5 (where deaths are delayed by a number of days) (Janke 2014); however, there is less understanding on longer-term lags. Hereafter, the analysis incorporates a lag of 1-time unit (1 month) to the response variable and assesses the model's performance.

**Table 1.** Linear Regression Model Results for Excess Mortality.

| Dependent Variable | Excess Mortality | | |
|---|---|---|---|
| | Linear Model | Linear Model AR(1) | Linear Model AR(1) with Lagged PM2.5 |
| | (1) | (2) | (3) |
| Maximum PM2.5 Concentration ($t$) | 0.001 *** | 0.001 *** | |
| | (0.0001) | (0.00005) | |
| Maximum PM2.5 Concentration ($t-1$) | | | 0.001 *** |
| | | | (0.00005) |
| Excess Death Rate ($t-1$) | | 0.415 *** | 0.434 *** |
| | | (0.002) | (0.002) |
| Low Temperature Index ($t$) | −0.002 *** | −0.001 *** | −0.001 *** |
| | (0.0003) | (0.0003) | (0.0003) |
| High Temperature Index ($t$) | −0.002 *** | −0.002 *** | −0.002 *** |
| | (0.0002) | (0.0002) | (0.0002) |
| Age Band ($t$) | 0.005 *** | 0.011 *** | 0.011 *** |
| | (0.0002) | (0.0002) | (0.0002) |
| Gender | 0.018 *** | 0.022 *** | 0.022 *** |
| | (0.002) | (0.002) | (0.002) |
| County | 0.0002 *** | 0.0003 *** | 0.0003 *** |
| | (0.00004) | (0.00003) | (0.00003) |
| Month and Year ($t$) | 0.00001 *** | 0.00000 *** | 0.00000 *** |
| | (0.00000) | (0.00000) | (0.00000) |
| Constant | −0.111 *** | −0.119 *** | −0.131 *** |
| | (0.009) | (0.009) | (0.009) |
| Observations | 459,248 | 459,248 | 457,236 |
| $R^2$ | 0.002 | 0.139 | 0.145 |
| Adjusted $R^2$ | 0.002 | 0.139 | 0.145 |
| RMSE | 0.7938 | 0.7374 | 0.7363 |
| Residual Std. Error | 0.794 (df = 459,240) | 0.737 (df = 459,239) | 0.736 (df = 457,227) |
| F Statistic | 144.894 *** (df = 7; 459,240) | 9265.396 *** (df = 8; 459,239) | 9710.495 *** (df = 8; 457,227) |

Note: The RMSE of a random walk model is 0.8571. *** $p < 0.01$.

Model residuals are clustered around the mean, and the normal Q-Q plot indicates other variables can be added to improve the fit. Based on the linear model, we can only conclude that PM2.5 concentrations are a significant explanatory variable.

The same regression is run using a generalised additive model (GAM). A GAM is a generalised linear model with "a linear predictor involving a sum of smooth functions of covariates" Wood (2017). GAM is appropriate when the relationship between variables is complex and provides a flexible non-linear relationship between the response variable and the predictor (Wood 2017). The response variable is the excess mortality rate as defined above, and we use similar predictors as per the linear model for comparability. We have compared three regressions using different explanatory variables:

Case 1: PM2.5 concentration at time ($t$) plus age, gender, county variables
Case 2: AR(1) with PM2.5 concentration at time ($t$) plus age, gender, and county variables
Case 3: AR(1) with PM2.5 concentration at time ($t - 1$) plus age, gender, and county variables

Results show monthly maximum PM2.5 concentrations are significant in explaining excess deaths, but the model's overall fit as measured by R-squared is poor. Consistent with the linear model, the best GAM regression is model (3) based on comparisons of RMSE. RMSE for all three regressions are included in Table 2, and GAM partial plots for model (3) are shown in Figure 5.

**Table 2.** Excess Mortality ($t$) RMSE comparison between LM and GAM.

| Dependent Variable | Excess Mortality | | |
|---|---|---|---|
| | **Case 1** | **Case 2: AR(1)** | **Case 3: AR(1) with Lagged PM2.5** |
| Linear Model | 0.7938 | 0.7374 | 0.7363 |
| GAM | 0.7916 | 0.7264 | 0.7264 |

Case 3 performs the best out of the linear and generalised additive models. Therefore, the partial plots for case (3) only are shown in Figure 5. The improvement in RMSE for the GAM regression between case (2) and (3) is marginal and less than 0.0001. Model diagnostics and additional partial plots are included in the Appendix A.

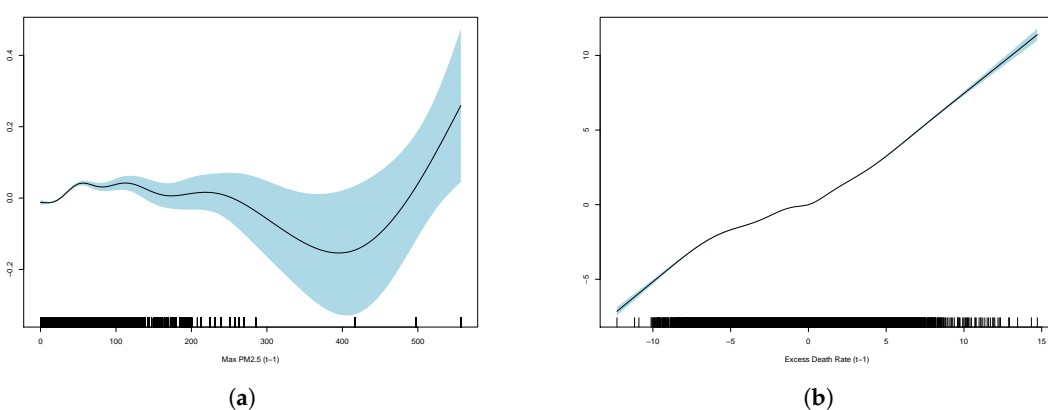

(**a**)            (**b**)

**Figure 5.** GAM Partial Plots. (**a**) GAM Case 3: PM2.5($t - 1$) Partial Plot. (**b**) GAM Case 3: Excess Death ($t - 1$) Partial Plot.

It is clear that the excess death ($t - 1$) factor has a close to a linear relationship with the excess death at time $t$.

To compare the linear and GAM models, we computed the AIC and R-squared. The R-squared of the GAM is almost four times higher than that of the linear model. However, it is still well below 0.1, indicating a poor fit. The RMSE of the first linear model (with PM2.5 at time $t$) is 0.7938, whereas the RMSE of the third linear model with lagged PM2.5 and lagged excess mortality is 0.7363. This suggests the linear model with lagged effects on excess mortality and PM2.5 performs slightly better. For comparison, the GAM RMSE of the first model with PM2.5($t$) is 0.7916 (an improvement from the linear model), whereas the RMSE of the third GAM model with lagged excess mortality and PM2.5($t - 1$) is 0.7264. This is consistent with the linear model results that suggest lags improve the overall model.

We have omitted the commentary of the second model as it is clear the model incorporating both lags performs the best. Finally, we also perform a $\chi^2$ test which further evidence the GAM is an improvement to the linear model. This is unsurprising as there are non-linear factors that contribute to excess deaths.

While the regression is not strong in either the linear model or GAM, the residuals are clustered towards the mean, and both regressions indicate that PM2.5 concentration is a significant factor. As previously stated, the GAM performs better than the linear model.

Finally, before comparing these results with the proposed method of a gradient boosting model, we further compute out-of-sample RMSE (using the same data split as the XGB Chen et al. 2022) and run a comparison of excess deaths using a simple random walk. The RMSE using a simple random walk is 0.8571, indicating a statistical model performs better.

A gradient boosting regression model, specifically, eXtreme gradient boosting ("XGBoost", "XGB") is the proposed model for forecasting. XGBoost is a regularised learning tool and uses the gradient boosting decision tree decision boosting algorithm. Boosting is a type of ensemble machine learning technique that uses the previous model's results as an input into the next one, with the new model being trained to correct the errors of the previous one. Models are added sequentially until no further improvements can be made; a gradient descent algorithm is used to minimise the loss when adding new models (Chen and Guestrin 2016). The unique feature of XGBoost is that it is often more efficient than other machine learning algorithms and enables scalability over large data sets. For reference and reproducability, the code used is available at: https://github.com/zm-dong/mortality_airpollution, accessed on 10 March 2022.

We now assess the results of the XGB. RMSE is used to optimise the model in selecting the boosting iterations that the model is run for training. Tree depth represents the number of decision nodes, and learning speed represents how much the next model corrects for the errors of the first model. RMSE is also the loss function used to compare the performance of the XGB model with GAM and linear regressions since we want to compare losses rather than the residuals. The R package "caret" (Kuhn 2022) was used to fit predictive models over different tuning parameters, an optimisation algorithm was run, and the best fit was selected based on RMSE.

Of the total data, 80% of the data were used for training, while the remaining 20% were used for testing. After the initial model construction, we tested the importance of variables and removed explanatory factors with poor explanatory power to improve the model. The model was tuned by adjusting the number of iterations used to train, the number of nodes, observations per node, and depth of trees, $\eta$ (the shrinkage or learning rate), and $\gamma$ (the minimum loss reduction, or pruning of trees).

The final model tuning parameters to give the lowest RMSE was: 500 repetitions, with a max tree depth of 4, $\eta = 0.025$ and $\gamma = 1$. The number of nodes per tree was also limited by setting a minimum number of samples in a node to 3.

Figure 6 shows RMSE for a subsection of the tuning tests for XGB. Based on the configurations shown, an increase in $\eta$ (the shrinkage or learning rate) or $\gamma$ (the minimum loss reduction or pruning of trees) does not improve the RMSE. We noted that running an XBG was also more efficient in terms of computational time and optimizing tuning parameters when compared with the generalised additive model.

The comparison of RMSE is shown for case 3 (AR(1) with a one-month lag in PM2.5) since the results and the LM and GAM regressions suggest this is the more optimal model. Table 3 shows that the XGB model has better predictive power based on the in-sample RMSE. The out-of-sample RMSE suggests that GAM performs better, whereas the in-sample RMSE indicates the XGB fits better. Therefore, the results suggest a classic example of overfitting when using a gradient boosting approach on this data set. While we can take further steps to reduce overfitting, the out-of-sample GAM result is better than the in-sample RMSE for XGB. Parameterising the XGB model is faster than that of GAM, which is consistent with the design of the XGB algorithm for the efficiency of computation time and memory resources.

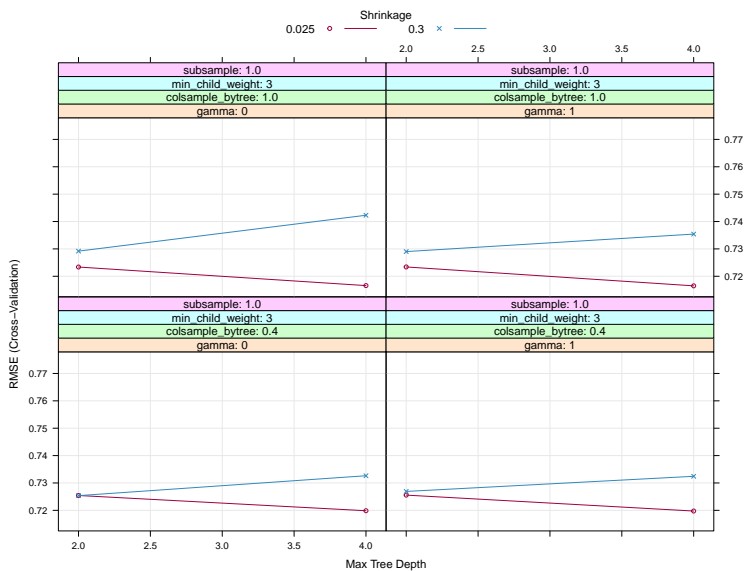

**Figure 6.** XGB Tuning Selections.

**Table 3.** Excess Mortality fit using XGB compared with LM and GAM—in-sample and out of sample RMSE.

| Dependent Variable | Excess Mortality (t) | | |
|---|---|---|---|
| | LM | GAM | XGB |
| | (1) | (2) | (3) |
| RMSE (in sample) | 0.7363 | 0.7264 | 0.7165 |
| RMSE (out of sample) | 0.7206 | 0.7100 | 0.7179 |

Understanding the importance of the explanatory variables in predicting mortality helps to provide more insight on whether PM2.5 concentration is a strong predictor of excess mortality. For example, if the variable "Year_Month" is relatively more important, trend factors over time are not captured by the explanatory variables. Similarly, if we see the geographic variable "County" is relatively more important, this suggests location factors impact mortality but are not captured by our data. The importance plot of this model in Figure 7 shows the prior period excess deaths is most important, though the Maximum PM2.5 is the second most important factor.

The importance plot is included in Figure 7 for the XGB model to understand the drivers of the forecast further. Even though the out-of-sample RMSE does not perform as well as GAM, the findings from the in-sample RMSE suggest the model is slightly stronger than the GAM. The importance plot indicates that adding PM2.5 concentration improves the accuracy of the prediction of excess mortality, but most of the prediction is driven by the historical excess mortality. Other important variables are temperature, time, and location factors. The importance of the location factor "County" suggests that other geographic variables in the data are not captured by age, gender, temperature, and PM2.5 which contribute to excess mortality as defined above.

The interesting observation is the lack of predictive power of wildfire events (as per the State of California Department of Forestry and Fire Protection). One explanation is that the result of a wildfire event is already reflected in the PM2.5 concentrations, and the event itself does not contribute to additional predictive power in determining deaths.

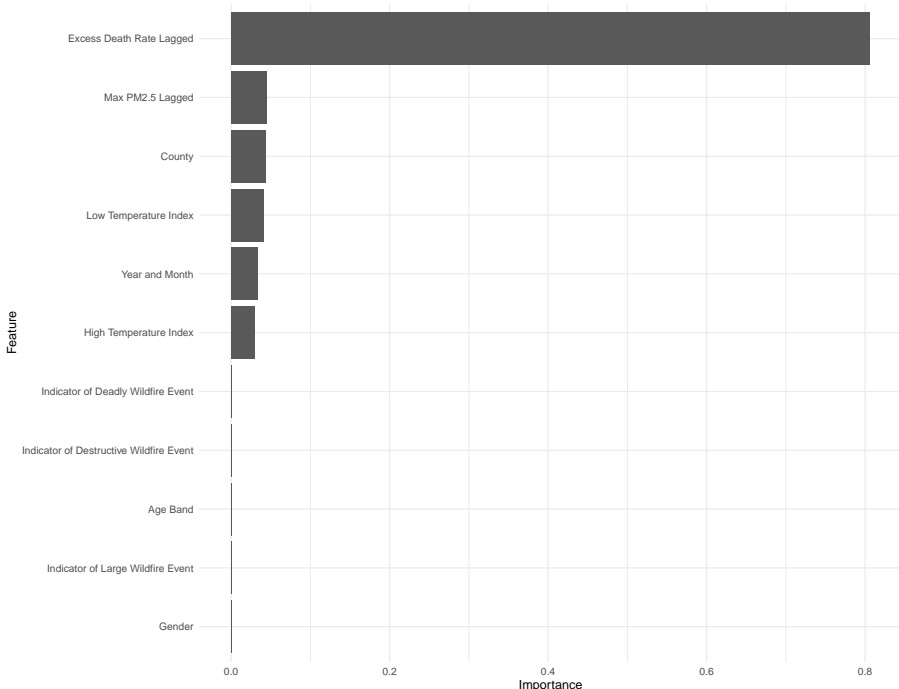

**Figure 7.** Feature Importance from XGB model.

## 4. Change Point Analysis

Results in Section 3 suggest a sophisticated machine learning approach such as XG-Boost does not necessarily perform better than more traditional statistical analysis models such as GAM to understand the impact of air pollution on mortality and forecast excess mortality. The findings indicate that none of the models which were tested have high predictive power as measured by RMSE. A potential explanation for the poor predictive power of machine learning techniques is the impact of structural breaks in the time series data introduced by extreme events (such as wildfires), which changes the underlying distribution of the data and results in low predictability. To better understand why the predictive power is poor, this section conducts change point analysis to identify structural breaks in excess deaths and compares this against structural breaks in PM2.5 concentration to explore the importance of extreme events. Using the detected structural breaks and information on historical Californian wildfires, we conclude that the impact of wildfires causes structural breaks in the model. Therefore, excess death experience is impacted by significant events such as wildfires.

To understand change points, the package "changepoint" (Killick and Eckley 2014) was used in ®️ (R Core Team 2022) on the existing data set. We also tested using the package "ecp" (James and Matteson 2014), specifically, E-Divisive, to perform hierarchical divisive estimation of multiple change points on multivariate data. (We can then compare this with E-Agglo, which estimates change periods using a priori assumptions about known change points—in this case, they would be the dates of wildfires impacting PM2.5).

Using the ecp package, two clusters were identified for excess mortality and three clusters for PM2.5 concentrations (i.e., two change points for excess mortality and three change points for PM2.5).

Break points in excess mortality were identified as:

- December 2010
- December 2017

There were no large wildfire events in California as per CalFire data CalFire (2022). However, the 2017 increase may be linked to the wildfires over October–December 2017 spanning six counties and identified as some of the largest and most destructive wildfires on the CalFire database (CalFire 2022).

Break points in maximum PM2.5 concentration were identified as:

- June and August 2008
- March 2010

As mentioned in the prior paragraph, there were no large wildfire events in California as per CalFire data CalFire (2022). However, it is interesting to see increases in PM2.5 and excess mortality identified in 2010 as a statistical breakpoint. The other breakpoints identified in 2008 correspond to the wildfires in Siskiyou, Monterey, and Trinity at the same time. CalFire classified these fires as some of the largest fires (CalFire 2022).

Surprisingly, the change points are increasing over time. Change points identified for PM2.5 and excess mortality do not correspond directly; however, this is likely due to the other factors driving excess mortality changes, as seen in the modeled results.

Therefore, while not all wildfires correspond to a breakpoint in terms of changes in excess mortality, they may trigger a breakpoint in the maximum PM2.5 concentrations observed.

## 5. Discussion and Key Findings

Our findings show an improvement in predictive power from incorporating PM2.5 as a factor in determining excess deaths. Importantly, we find that adding PM2.5 as a factor produces a stronger outcome than a random walk and that incorporating lags to PM2.5 concentration marginally improves the prediction.

In terms of methodology, the results suggest a generalised additive model performs the best in terms of out-of-sample RMSE. While the machine learning method (XGB) performs better in-sample based on the data set used, there is evidence of overfitting the data.

Changepoint analysis suggests there are change points in excess mortality driven by changes in PM2.5. However, not all breakpoints for PM2.5 concentrations correspond to a breakpoint in excess deaths.

It is challenging to understand the effect of PM2.5 on mortality given the interactions between variables, consideration for break points, and availability of data. The implications of the statistical results conducted in this paper suggests there is indeed an impact on mortality from increases in PM2.5. Among all the models we compared, GAM performs the best and is the easiest in terms of computation. For larger sample sizes, it may be advantageous to apply XGB instead.

*Limitations and Future Works*

As shown in the results, the overall R-square and performance of the models are not high. The findings of this paper largely depend on the data that was used. While we can compare different methods and conclude that a GAM performs the best in out-of-sample forecasts, further improvements in predictive power can be made with a stronger data set.

It is also worth considering the metric used as the baseline to measure excess mortality. Nepomuceno et al. (2022) explores the sensitivity of excess mortality from a COVID-19 lens and finds excess mortality rates vary substantially with changes in the reference period, mortality index, method, and time unit of death. Therefore, further comparison can be explored in this context of air pollution to understand the predicted impact.

To better understand the predicted mortality impacts, interval forecasts will provide a better view of model uncertainty. The analysis in this paper only focuses on the central estimate. With improved data, it would be valuable to understand further the distribution of increased excess mortality in future works.

Other improvements to the analysis include:

- Considerations for spatial dependence: in this analysis, the county factor represents other geographic features not captured by PM2.5. However, some dependencies are not captured using this approach and are important to consider in the context of climate events.
- Adjustments for life-years exposed to improve the understanding of excess deaths by age

- - Distributed time-series models with the inclusion of higher-order lags and the use of extreme value theory

Given the prevalence and magnitude of risks that ambient air pollution presents, it is surprising that the actuarial literature on air pollution is limited. The historical insurance industry lens to understand climate risk factors has been general insurance, such as impacts of floods and cyclones. The implications from a life insurance and mortality perspective are less well understood due to the complexities of long-term exposure effects, interactions with co-morbidities and other risk factors, direct and indirect impacts, and data limitations. As climate change and extreme weather events occur with increasing frequency and severity, life insurers will need to understand better the nature of risks and the changing landscape of these exposures to risk.

Globally, regulators have emphasised the importance of risk management and the quantification of climate risks. In 2020, the European Insurance and Occupational Pensions Authority (EIOPA) issued an Opinion on the supervision of the use of climate change risk scenarios in Own Risk and Solvency Assessment (ORSA), a compulsory capital disclosure for life insurers (EIOPA 2021). Similarly, the Bank of England (BoE) has published guidelines around climate-related financial disclosures, in which the Bank issues its support for the Task Force on Climate-related Financial Disclosures (TCFD) (Bank of England 2020). The Australian Prudential Regulation Authority (APRA) has also declared that climate-related risks are "distinctly financial in nature… foreseeable, material, and actionable now" (The Australian Prudential Regulation Authority 2021). APRA has since released guidelines to insurers on managing the financial risks of climate change in response to industry requests for greater clarity of regulatory expectations (The Australian Prudential Regulation Authority 2021). In 2021, one of APRA's cross-industry priorities and strategic focuses is to conduct a climate change supervisory review.

Historical climate events that define "extremes" show the impacts of climate change are global. These actions by financial and insurance industry regulators further evidence the growing recognition that climate factors play a role in determining the risks to human lives and impacts upon insurance risks. Therefore, a better understanding of the impacts of air pollution from climate events is required to quantify the risks that insurers are exposed to, regardless of location.

All code is available from https://github.com/zm-dong/mortality_airpollution, accessed on 10 March 2022.

**Author Contributions:** Conceptualization, Z.M.D., H.L.S. and A.B.; methodology, Z.M.D., H.L.S. and A.B.; investigation, Z.M.D., H.L.S. and A.B.; resources, Z.M.D.; data curation, Z.M.D.; writing—original draft preparation, Z.M.D.; writing—review and editing, Z.M.D., H.L.S. and A.B.; visualization, Z.M.D.; supervision, H.L.S. and A.B.; projection administration, H.L.S. All authors have read and agreed to the published version of the manuscript.

**Funding:** This research received no external funding.

**Data Availability Statement:** All the datasets used in this study are available from the links provided in the manuscript.

**Acknowledgments:** The authors acknowledge insightful comments and suggestions provided by seminar participants, particularly Professors Edward Frees, Tim Higgins, and Francis Hui, at the Research School of Finance, Actuarial Studies and Statistics, Australian National University.

**Conflicts of Interest:** The authors declare no conflict of interest.

## Appendix A. Model Components and Residual Diagnostics of the Selected GAM Regression

These are partial plots of the GAM regression for Model 3. We have also included the degrees of freedom, *F*-statistic, and *p*-values for the parametric and non-parametric components of GAM.

Further, the model diagnostics for GAM are included, which shows the residuals having a broadly "cloud" formation. The residuals are clustered around 0. However, the Q-Q plot suggests other variables can be added to the model to improve the performance. This is consistent with our findings.

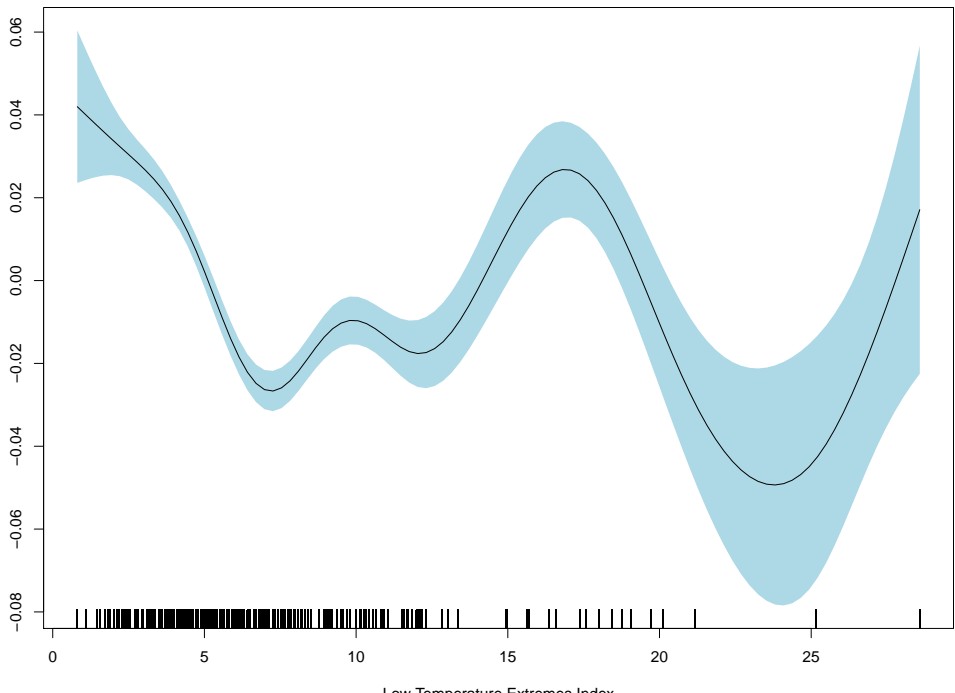

**Figure A1.** GAM Fit (3).

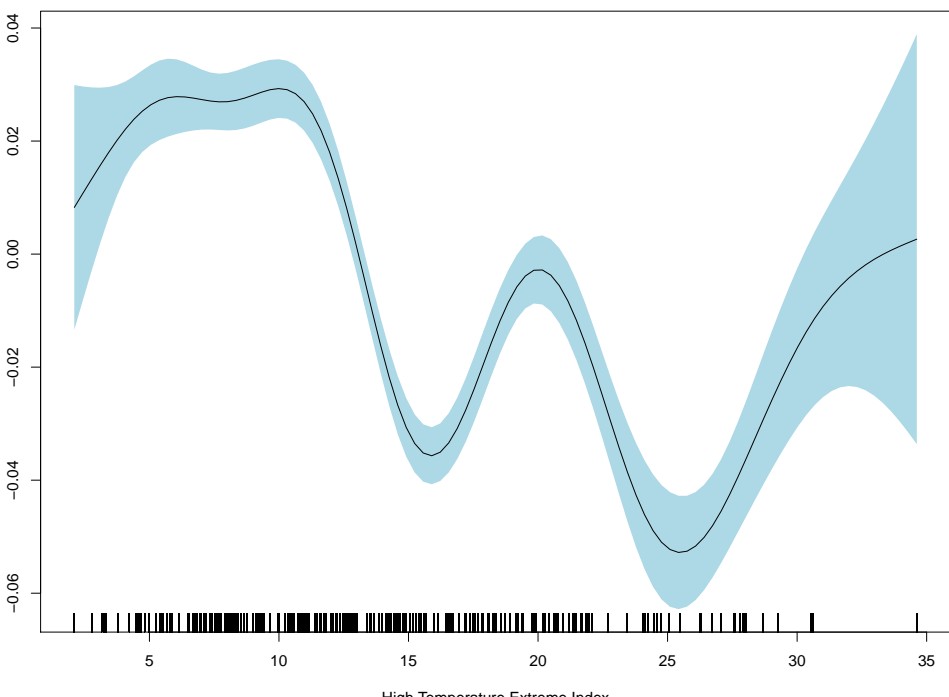

**Figure A2.** GAM Fit (4).

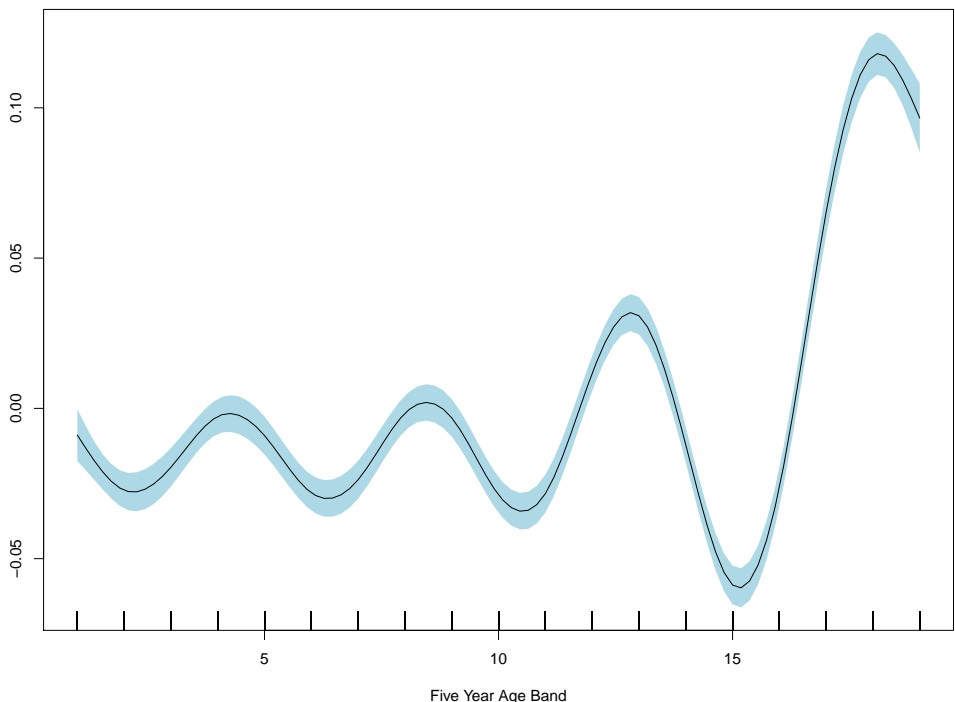

**Figure A3.** GAM Model Fit (5).

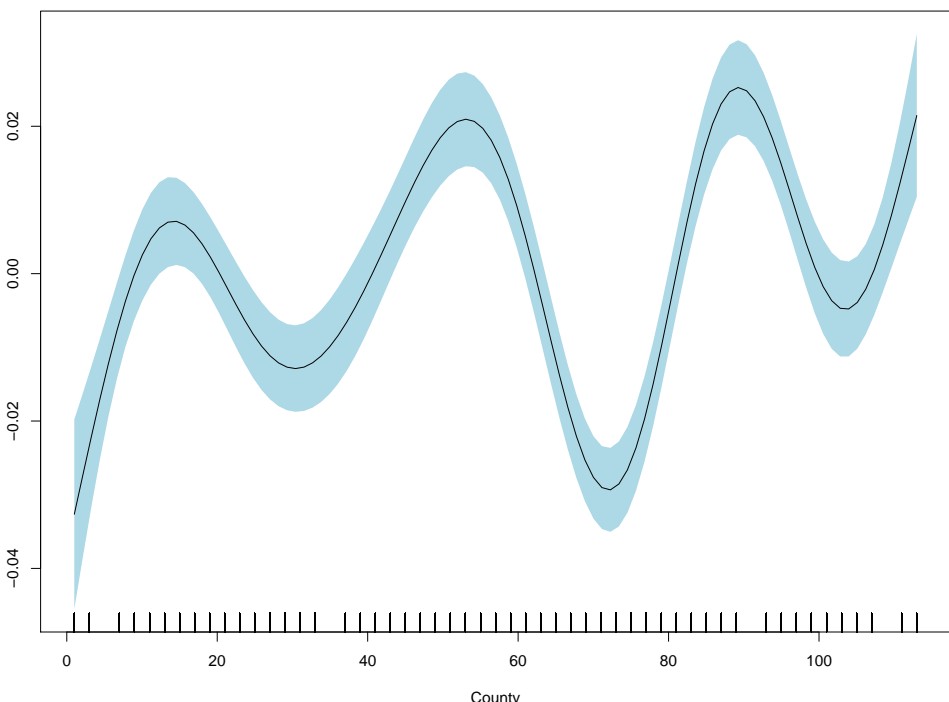

**Figure A4.** GAM Model Fit (6).

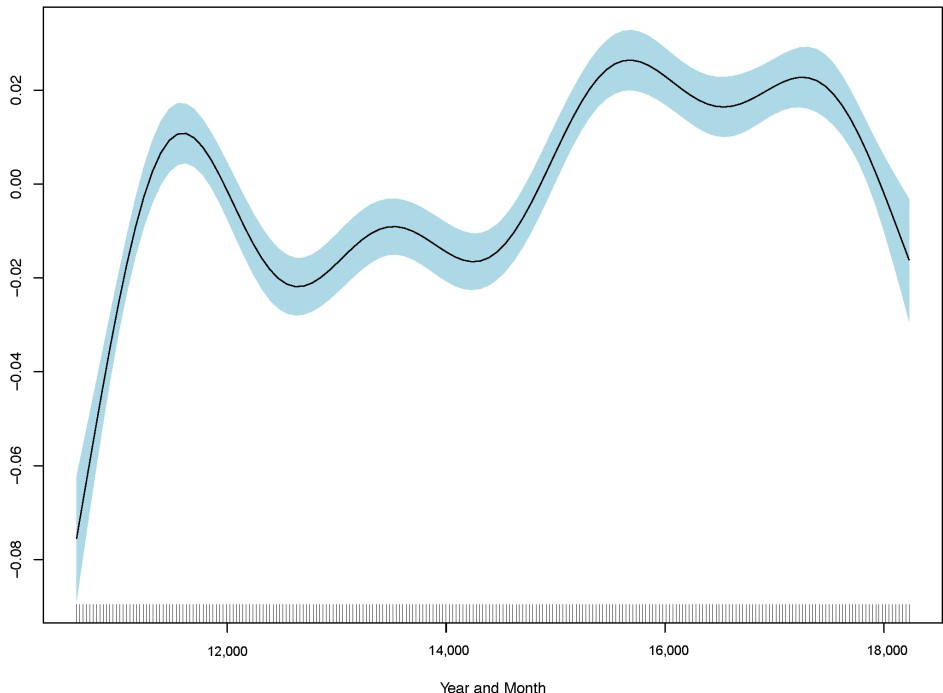

**Figure A5.** GAM Model Fit (7).

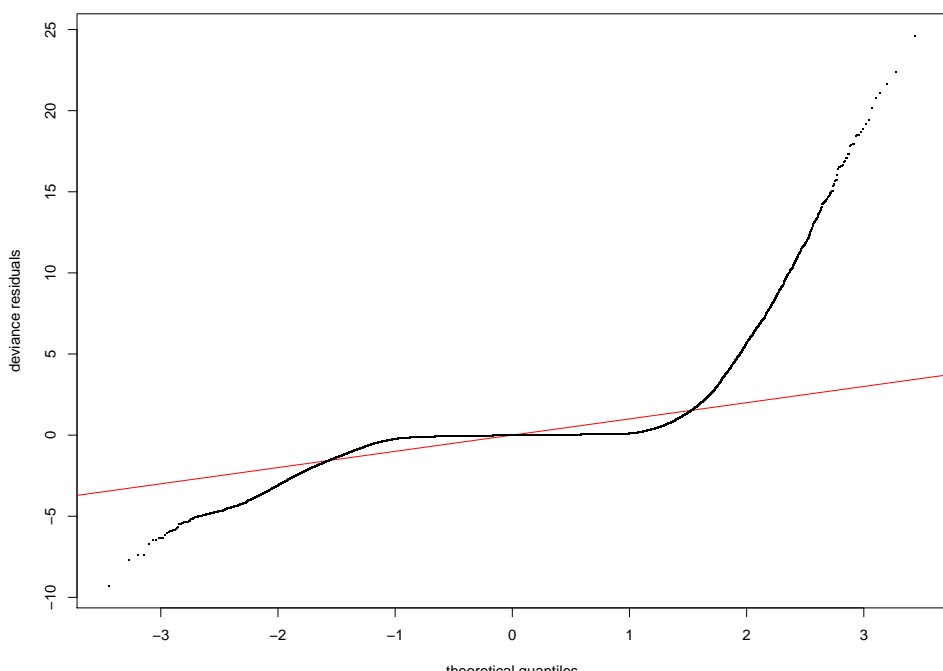

**Figure A6.** GAM Model Diagnostics (1).

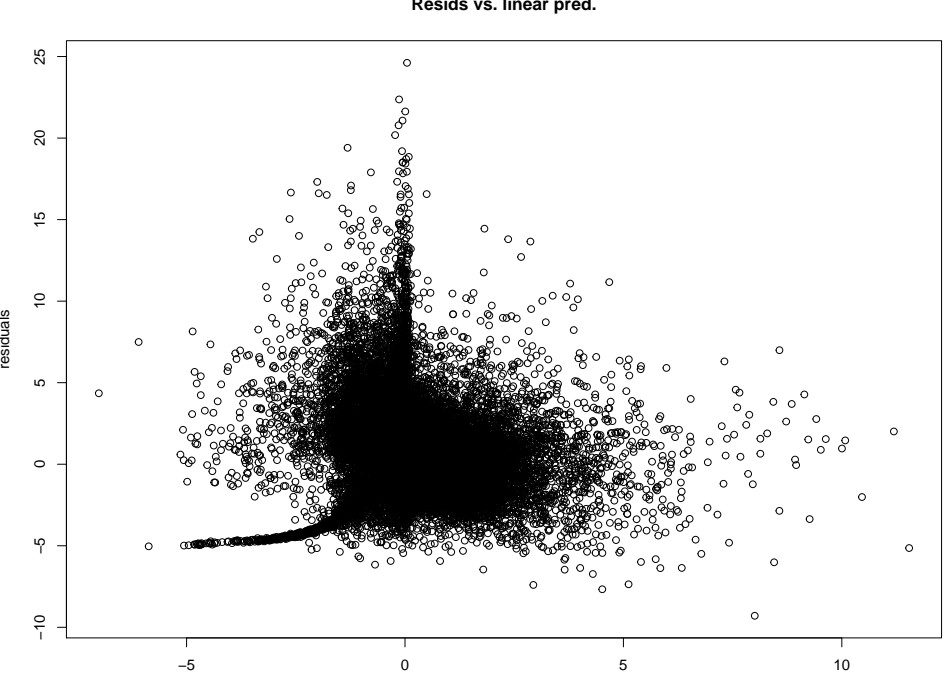

**Figure A7.** GAM Model Diagnostics (2).

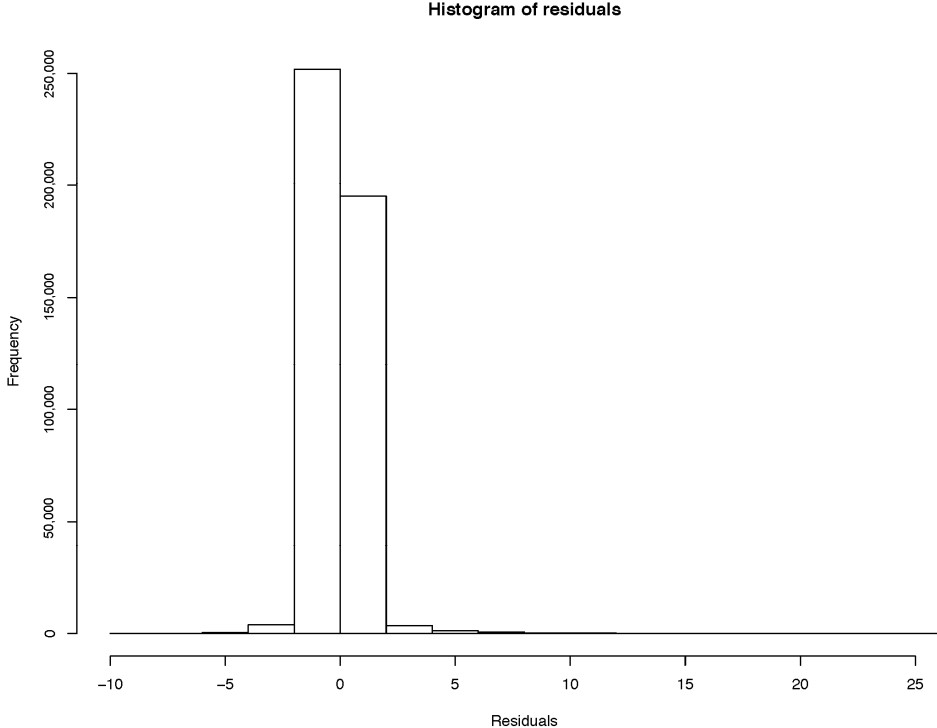

**Figure A8.** GAM Model Diagnostics (3).

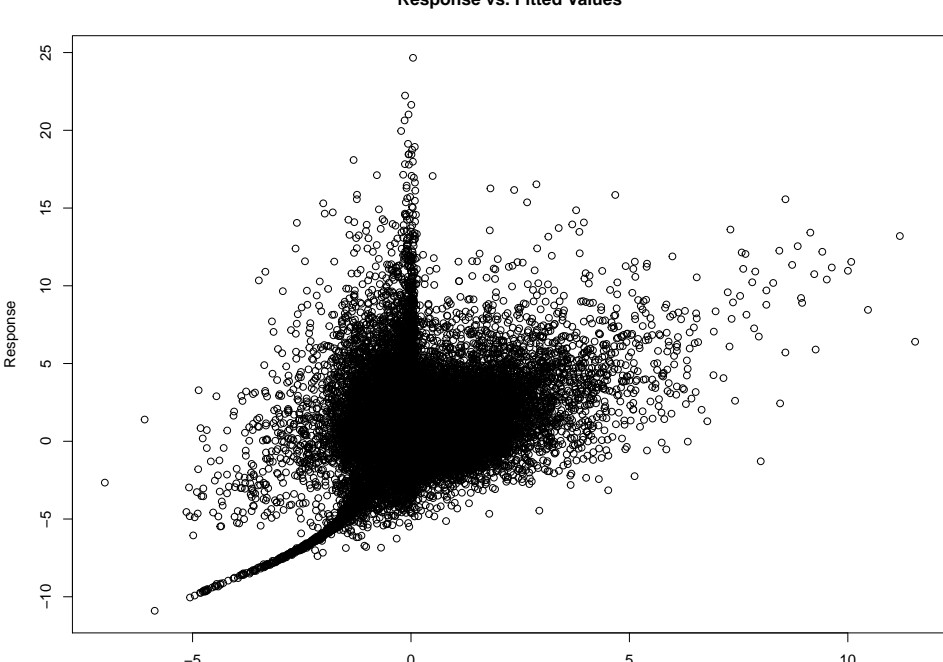

**Figure A9.** GAM Model Diagnostics (4).

**Table A1.** Excess Mortality fit using GAM: estimated non-parametric components, with the corresponding effective degrees of freedom, *F*-statistic, and *p*-value.

| Statistic | N | Mean | St. Dev. | Min | Pctl(25) | Pctl(75) | Max |
|-----------|---|------|----------|-----|----------|----------|-----|
| edf | 5 | 8.647 | 0.364 | 8.222 | 8.282 | 8.885 | 8.977 |
| Ref.df | 5 | 8.937 | 0.083 | 8.833 | 8.861 | 8.995 | 9.000 |
| F | 5 | 64.004 | 71.046 | 6.401 | 21.025 | 54.078 | 185.668 |
| *p*-value | 5 | 0.000 | 0.000 | 0 | 0 | 0 | 0 |

**Table A2.** Excess Mortality fit using GAM and autoregressive factors—parametric components.

| Statistic | N | Mean | St. Dev. | Min | Pctl(25) | Pctl(75) | Max |
|-----------|---|------|----------|-----|----------|----------|-----|
| Estimate | 3 | 0.004 | 0.011 | −0.005 | −0.002 | 0.008 | 0.016 |
| Std. Error | 3 | 0.001 | 0.001 | 0.00004 | 0.001 | 0.002 | 0.002 |
| t value | 3 | 5.616 | 7.815 | −3.098 | 2.421 | 9.972 | 12.004 |
| Pr(>|t|) | 3 | 0.001 | 0.001 | 0.000 | 0.000 | 0.001 | 0.002 |

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
