# Peer review of "Air Pollution and Mortality Impacts"

_risks, doi:10.3390/risks10060126_

Round 1

Reviewer 1 Report

Dear Authors,

I am sending comments to the article " Air Pollution and Mortality Impacts ".

Very well written article. basically I have no comments on the article. In my opinion, the authors should point out that state policy has a serious impact on the level of air pollution. for example, in Poland, air pollution is the highest in Europe. Poland should give up coal, mines are unprofitable and coal poisons the air. However, the government is afraid to close unprofitable mines because it may lose votes in the elections.

Zimon, G.; Zimon, D. The Impact of Purchasing Group on the Profitability of Companies Operating in the Renewable Energy Sector—The Case of Poland. Energies 2020, 13, 6588. https://doi.org/10.3390/en13246588

Unfortunately, there is no chance for changes in Poland, it seems to me that it is similar in Eastern and Central Europe.

The article must clearly state that government policy has a large impact on the level of air pollution.

Sincerely

Author Response

Please refer to the attached response_letter.pdf. Thank you.

Reviewer 2 Report

This paper deals with assessing the excess of mortality taking into account the air quality impact on population. The authors consuct an empirical study using monthly Californian climate and mortality data from 1999 to 2019 to determine whether adding PM2.5 as a factor improves forecast excess mortality. 

This paper is well written and structured. Both the quality of presentation and the scientific soundness are high.

Some suggestions to the authors:

  1. In the abstract section you refer to  "an actuarial perspective". Please, explain to the reader this statement;
  2. Explain to the reader what PM2.5 is;
  3. Give a brief description of GML, GAM and XGB models;
  4. I suggest to expand concluding section giving more details about your results.

Author Response

(The authors gave the same response as above.)

Reviewer 3 Report

Comments on the manuscript

“Air Pollution and Mortality Impacts”

This paper uses monthly Californian climate data and the mortality data from 1999 to 2019 to investigate the research question whether PM2.5 has significant impact on mortality. The authors use excess mortality rate as the response variable and apply three different regression models, a Generalized Linear Model (GLM), a Generalized Additive Model (GAM), and an Extreme Gradient Boosting (XGB) regression model, to examine the impact of PM 2.5 on mortality, by comparing the predictive power after incorporating (1) the maximum PM2.5 Concentration in the current period, (2) the maximum PM2.5 Concentration in the current period and Excess Death Rate in the previous period, and (3) maximum PM2.5 Concentration and Excess Death Rate both in the previous period, into the regression models. Their research contributes to the current literature and provides useful information to insurance practitioners, researchers, policymakers, and other stakeholders. The paper is acceptable with minor revision. The following comments on either technical or stylistic contents of the paper are given to improve reader experience and increase the impact of the paper.

Comments on content

1.     Page 5, Line 213, The authors state that “the index was developed using 1960 - 1990 as reference years, and subsequent reporting is based on this standardized timeframe.” Then, how does the index was in line with other data whose time frame is from 1999 to 2019. A little bit more explanation would be helpful.

2.     Page 7, Line 260. A formula for defining excess mortality rate would be useful for readers to understand the concept and how it is computed based on death counts and population size, which are from different data sources.

3.     Page 7, Line 264-265. The authors state that “Preliminary analysis shows that average and minimum PM2.5 concentrations are less significant than maximum PM2.5 concentration readings.” This statement is in conflict with the conclusion “Other regressions using monthly average and monthly minimum PM2.5 concentrations were run; results were similar to the regression using maximum PM2.5 concentration readings…” on Page 8, Line 285-287.

4.     Page 7, Line 252. Formulation of three models at the beginning of Section 3 would be helpful for readers to understand the methodology. The authors are experts of the methodology; however, readers might not have such background.

5.     Page 8, Line 271. It would be appreciated to give more interpretation about the numbers in Table 1 to explain the evidence that supports the conclusion “that maximum PM2.5 concentrations are a significant factor in explaining excess deaths”. There is no information of p-value and the hypothesis test behind the numbers.

6.     Page 12, Line 399. How to define and quantify predictive power using RMSE?

Stylistic comments

The reviewer strongly suggests professional English editing to improve the impact of the paper.

1.     Here are some typical examples of confusing writings in the Introduction.

(1)    Page 1, Line 39 - 42, the sentence “There has been…” is confusing and the reviewer couldn’t understand the grammar in this sentence  

(2)   Page 1, Line 42 – 43,  “ Recent wildfires in the US have been worse than historical fires, research suggesting 42 this is also an impact of climate change” . Does the authors means “There are more disastrous wildfires in the US than before, probably due to climate change.” ?

(3)   Page 1, Line 45- 47,   “Thus, there is a compelling need to understand better the impacts on air pollution mortality triggered  by extreme events.”  It might be better to rephrase the sentence as “…better understand the impact on mortality of air pollution triggered by extreme natural disasters.”

(4)   Page 2. Line 98- 100, “ for example, Huang et al. (2019) performed a cohort study based in China to understand the impact of PM2.5 on stroke over 15 a period of years from 2000 to 2015 (Huang et al. 2019).”

The reviewer cannot list many other phrases that need to be corrected and/or clarified in Introduction section. 

2.     Page 4, Line 170, what does “rolling 5-year average” mean?

3.     Page 5, line 191, it would be appreciated if a URL link or reference is provided for the data source.

4.     Page 5, Line 198-202, what is the PM2.5 concentration is aligned with when averaged across all counties?  Figure 2 is for one county. The reviewer is confused about the information the authors are trying to deliver here.

5.     Page 8, Table 1, the caption of the table “Excess Mortality fit using..” might better be changed to “Excess Mortality fitted by…”, same to the caption of Table 2, 3, 4, and 5.

6.     Page 8, Line 268. What does “RMSE” stand for? A full name needs be given for the abbreviation for the first time of use in the context.

7.     Page 9, Line 297-300. Model 1, Model 2, and Model 3 listed here are confusing with the three regression models, when the authors mentioned the models in the following context. Therefore, it might be better to use Case 1, Case 2, and Case 3. For example, readers might not understand what “the models” are in the sentence “The findings indicate that none of the models which were tested have high predictive power as measured by RMSE” on Page 12, Line 399.

Author Response

(The authors gave the same response as above.)
